# The Effect of DBD Plasma Activation Time on the Dyeability of Woven Polyester Fabric with Disperse Dye

**DOI:** 10.3390/polym13091434

**Published:** 2021-04-29

**Authors:** Thu Nguyen Thi Kim, Khanh Vu Thi Hong, Nguyen Vu Thi, Hai Vu Manh

**Affiliations:** 1School of Textile—Leather and Fashion (STLF)—Hanoi University of Science and Technology (HUST), No. 1, Dai Co Viet, Hai Ba Trung, Hanoi 11615, Vietnam; thu.nguyenthikim@hust.edu.vn (T.N.T.K.); hai.vumanh@hust.edu.vn (H.V.M.); 2Hanoi Industrial Textile Garment University, Le Chi, Gia Lam, Hanoi 12411, Vietnam; Nuyenvt@hict.edu.vn

**Keywords:** DBD plasma, polyester fabric, disperse dyes, dyeability, short plasma treatment time

## Abstract

This study consists of two parts. In the first, the woven polyester fabric, after washing to remove lubricant oils, was treated with the dielectric barrier discharge (DBD) plasma at the short plasma exposure time (from 15 to 90 s). The effect of the plasma exposure time on the activation of the polyester fabric was assessed by the wicking height of the samples. The results show that the wicking height in the warp direction of the plasma-treated samples improved but was virtually unchanged in the weft direction. Meanwhile, although the tensile strength in the warp direction of the fabric was virtually unaffected despite the plasma treatment time up to 90 s, in the weft direction it increased slightly with the plasma treatment time. Scanning Electron Microscope (SEM) images and the X-ray Photoelectron Spectroscopy (XPS) spectra of the samples before and after the plasma treatment were used to explain the nature of these phenomena. Based on the results of the first part, in the second part, two levels of the plasma treatment time (30 and 60 s) were selected to study their effect on the polyester fabric dyeability with disperse dyes. The color strength (K/S) values of the dyed samples were used to evaluate the dyeability of the fabric. The SEM images of the dyed samples also showed the difference in the dyeability between the plasma-treated and untreated samples. A new feature of this study is the DBD plasma treatment condition for polyester fabrics. The first is the use of DBD plasma in air (no addition of gas). Second is the very short plasma treatment time (only 15 to 90 s); this condition will be very favorable for the deployment on an industrial scale.

## 1. Introduction

Polyester is the most widely used fiber in the world due to its excellent mechanical and physical properties [1]. In 2018, polyester production accounted for approximately 55% of global textile fiber production [2]. However, polyester fabric has undesirable properties such as pilling, static and lack of dyeability [3]. These limitations are due to the fact that polyester is chemically inert, has a high crystal structure and small amorphous region [4,5]. Therefore, the dyeing of polyester fabrics needs to be done with synthetic disperse dyes. Disperse dyes are applied to polyester fibers by stable dispersing in water contain auxiliaries, especially dispersants at a high temperature (approximately 130 °C) and high pressure. Under these conditions, the amorphous region in the fiber swells, and the dyes are adsorbed on the fiber surface and then transferred into the fiber [4,5,6,7]. This process requires a lot of energy and special equipment. Carrier dyeing has been widely deployed as a method of reducing dyeing temperature for polyester by the changes in its physical properties, notably such as the glass transition temperature (Tg). However, carriers have significant problems such as: toxicity, unpleasant odors, sensitivity to human skin, environmental pollution and adversely affecting the physical properties of fibers [5,7]. Several methods have been investigated to improve the dyeing properties of polyester fabrics, including the surface modification methods. The hydrolysis of polyester fibers with an alkaline solution such as sodium hydroxide is one of the most traditional methods. During alkaline hydrolysis, polyester undergoes nucleophilic replacement and chain scission of polyester occurs, resulting in a significant weight loss and formation of hydroxyl and carboxylate end groups. This helps to improve the moisture content and the dyeability of the polyester [8]. However, the weight loss, the mechanical strength loss of the polyester fabric and environmental problems are the limitations of this method [9,10]. Other alternatives to surface modification of polyester fabrics have been studied, such as: the use of enzymes [3], “laser-dye” [11], nanotechnology, electrochemistry, supercritical carbon dioxide dyeing, plasma, ultrasonic and microwave [5]. Among the various surface modification techniques, the plasma treatment seems to be well-suited to polyester fabrics. Plasma technology is based on a simple physical principle. When energy is supplied to matter, it changes state from solids to liquids, and liquids to gaseous. If more energy is supplied to a gas, it becomes ionized and goes into the energy-rich plasma state [12]. Plasma consists of free electrons, radicals, ions, atoms, molecules and different excited particles. It is the interaction between these excited particles with the solid surfaces placed in a plasma space, leading to the chemical and physical modifications of the material surface without damaging the bulk properties of the material [13,14]. In polyester fabric dyeing, with the help of plasma technology, it can save energy, water, time, dyestuff and auxiliaries. This leads to a reduction in the amount of wastewater pollution compared with conventional chemical treatment techniques [15]. Hence, plasma can be applied to activate the surface of polyester fabrics prior to dyeing to enhance their dyeability [4,16,17,18,19]. Except for study [19], the aforementioned studies all used the DBD plasma. DBD plasma is an atmospheric plasma that operates at a high voltage source. Having a dielectric layer, covering one or both electrodes, thanks to which the plasma in the DBD is cold, it is very suitable for textiles [13,20]. The studies [4,16,17,18] show that polyester fabrics treated with DBD plasma already had a higher wettability, so their color depth and their color fastness were also improved. Regarding the tensile strength of the fabric, study [17] found that there was no significant difference between the untreated and the DBD plasma-treated fabric. However, in the aforementioned studies, the plasma treatment time for the samples was relatively long (15 min in the study [4], 30 min in [16] and 7 min in [17]). Furthermore, several gases were used in plasma treatment (ammonia for study [4] and oxygen for study [16]). Thus, although the activation of polyester fabrics with DBD plasma prior to dyeing was quite effective, too-long plasma treatment times can become a weakness if these processes are deployed industrially. Therefore, it is necessary to investigate the dyeability of polyester fabric with a short plasma treatment time. For this reason, in this study, the DBD plasma was applied on the polyester fabric for a short time (from 15 to 90 s). The wicking height, the tensile strength, the SEM images and the XPS spectra of the samples before and after plasma treatment were measured to evaluate the effects of the short-time DBD treatment on the activation and the surface morphology of polyester fiber. Studies on dyeing of the untreated and plasma-treated polyester fabric with disperse dyes were also performed. The purpose of this study is to clarify the possibility of using a DBD plasma in a short time to increase efficiency in dyeing of the polyester fabric with disperse dyes. A new feature of this study is the DBD plasma treatment condition for the polyester fabric. The first is the use of the DBD plasma in air (no addition of gas). Second is the very short plasma treatment time (only from 15 to 90 s).

## 2. Materials and Methods

### 2.1. Materials 

Fabric: The 100 % polyester fabric, washed to remove lubricant oils, was supplied by NATEXCO-Vietnam. It was 1/1 plain weave fabric, composed of spun yarns with the construction of 18.5 × 2 × 18.5 × 2 (Tex)/59 × 44 (per inch) weighing 178 g/m^2^.

### 2.2. Methods

#### 2.2.1. Plasma Treatment Process for Polyester Fabric

The DBD plasma treatment for polyester fabric was carried out in the laboratory DBD plasma equipment (Figure 1) developed by the School of Engineering Physics (SEP) of HUST for the KC.02.13/16-20 project. The atmospheric pressure DBD plasma consisted of two parallel electrodes, each 8 × 50 cm^2^. Both electrodes were covered with a polycarbonate sheet as a dielectric layer. The thicknesses of the lower and upper sheets were 3 and 5 mm, respectively. The electrodes were cooled by a circulated oil system which was operated by a pump. The distance between the electrodes was adjusted at 3 mm. The fabric with a width of less than 50 cm in a uniform tension could move continuously between two parallel electrodes. In this study, the plasma power of 400 W (1 W/cm^2^) was fixed, while the plasma exposure time for the samples was changed from 15 to 90 s (15, 30, 45, 60, 75 and 90 s). Corresponding to the plasma treatment time, the plasma-treated samples were named P15, P30, P45, P60, P75 and P90 while the untreated sample was named P0.

#### 2.2.2. Characterization of the Control and Plasma-Treated Polyester Fabric

Capillary measurement

The capillary of the untreated and treated samples was assessed through their wicking height at standard atmosphere (temperature of 20 ± 2 °C and relative humidity of 65 ± 4%). The wicking height of the fabric was measured according to the TCVN 5073—90 Standard. Specimens with the sizes of 200 mm × 25 mm were prepared (3 strips in the warp direction and 3 in the weft direction of the fabric). The samples were conditioned in the standard atmosphere for 24 h before testing. The conditioned specimens were suspended vertically so that their bottom ends were dipped in a reservoir of Kali dicromat solution (1 g/l in the distilled water). The height of the fabric immersed in Kali dicromat solution was 10 mm. The height of rise of the liquid at the center of the strip above the water level was measured as a function of time every 5 min until 30 min. The final result was the average value of 3 specimens. The experiments were conducted in the standard atmosphere.

Tensile strength test

The tensile strength of the untreated and plasma-treated samples was determined as the maximum force (P_MAX_) in the tensile test. P_MAX_ was recorded when a test specimen was taken to rupture during a tensile test. The tensile test was carried out according to the ISO 13934–1: 2013 standard by Tenso Lab 2512A, Mesdan tensile testing machine (STLF, Hanoi, Vietnam)

Samples were prepared according to the Annex A2 of the ISO 13934–1: 2013 standard (three samples in warp direction and three in weft direction). The width and length of all test pieces were 50 ± 0.5 mm and 300 ± 0.5 mm, respectively. They were prepared according to Section 8.2 of the ISO 13934–1: 2013 standard. The gauge length of the tensile-testing machine was 200 ± 1 mm. The rate of extension of the tensile-testing machine was set to 100 mm/min.

Scanning Electron Microscope (SEM) images

SEM images of the untreated and plasma-treated samples were taken to clarify the surface change of polyester fibers resulting from the plasma treatment. 

FESEM: JEOL JSM-7600F at the Laboratory of Electron Microscopy and Microanalysis (BKEMMA) (Advanced Institute for Science and Technology (AIST), HUST, Hanoi, Vietnam) was used for these tests. The condition of SEM was at U = 5 kV, X (magnification) = 5000, 20,000, 50,000 and 100,000. All the samples were coated with platinum prior to observation by SEM.

X-ray Photoelectron Spectroscopy (XPS) measurement

The new functional groups of the polyester due to the plasma treatment were evaluated by XPS system. The Kratos Axis-Ultra DLD, Kratos Analytical (Shinshu University, Matsumoto, Japan) was used to verify the changes in the surface chemical composition of the samples before and after plasma treatment. The pressure in the XPS chamber was reduced to 6 × 10^9^ Torr before samples were excited with monochromatic Al K_α_ 1,2 radiation at 1.4866 KeV.

#### 2.2.3. Dyeing Procedure for the Untreated and Plasma-Treated Polyester Fabric

Based on the results of the above studies, in this study, 2 plasma treatment times (30 and 60 s) were selected to study the effect of the plasma treatment time on the dyeability of polyester fabric. 

Three types of samples (untreated polyester fabric (P0), polyester fabric treated with plasma for 30 s (P30) and polyester fabric treated with plasma for 60 s (P60)) were dyed with disperse dye Dianix Red S-G. Each type of fabric was dyed with 7 different dye baths containing varied concentrations of dyestuff (M) (M = 0.1, 0.5, 1.0, 1.5, 2.0, 3.0 and 4.5% owf shade). The liquor ratio was 1:10 and pH of the dye bath was adjusted to 4.5 by acetic acid for all dyeing. The polyester fabric was introduced into the dye bath at room temperature and then the temperature of dye bath was raised at a rate of 2 °C per minute to 130 °C. At this temperature, the dyeing was continued for 60 min. After that, the temperature of the dyeing bath was rapidly reduced in approximately 15 min. The dyed fabrics were then rinsed under running water until the water was colorless (visual evaluation) and dried at room temperature. Corresponding to the plasma treatment time and dye concentration of the dye bath, the dyed samples were named PD0-0.1, PD0-0.5, PD0-1.0, PD0-1.5, PD0-2.0, PD0-3.0, PD0-4.5, PD30-0.1, PD30-0.5, PD30-1.0, PD30-1.5, PD30-2.0, PD30-3.0, PD30-4.5, PD60-0.1, PD60-0.5, PD60-1.0, PD60-1.5, PD60-2.0, PD60-3.0 and PD60-4.5. All the experiments were carried out in the IR Lab dyeing machine Orintex (STLF, Hanoi, Vietnam).

#### 2.2.4. Rinsing the Dyed Samples with Acetone

After the aforementioned dyeing process, dyed samples were rinsed with acetone until no color remained in the acetone. This step was performed as follows: the 5 × 5 cm dyed sample was placed in the beaker of Washing Fastness Tester (STLF, Hanoi Vietnam), then 50 mL of acetone was poured into each beaker. The cup was closed tightly and the machine was ran at a constant speed of 40 rpm (+/− 2 rpm) at room temperature. Each rinse cycle lasted for 10 min. Observing the color of the acetone after each cycle, if the acetone in the beaker was still red, the acetone was replaced with a new beaker and the rinse cycle repeated until the acetone was no longer colored. The colors of the samples before and after acetone rinse were compared to confirm the dye absorbency of the samples.

Similar to the 3 sets of dyed samples, 3 sets of dyed samples rinsed with acetone were denoted PD0′, PD30′ and PD60′.

#### 2.2.5. Color Measurement

Color strength (K/S) measurement

The colors of the dyed and acetone-rinsed samples are expressed as their maximum K/S values, determined as follows.

The reflectance (R) of the samples from 400 to 700 nm was measured by Ci4200 SpectroPhotometer (STLF, Hanoi, Vietnam) with D65 illuminant and 10° standard observer. The color strength (K/S value) was then established according to the following Kubelka–Munk Equation (1) [4,16,17,18,19]
(1)$$K/S\;=\;\frac{{\left(1-R\right)}^{2}}{2R}=\frac{\mathrm{Ac}}{S}$$
where
K absorption coefficient;S scattering coefficient;R reflection factor.

c is the concentration of the absorbing species and A is the absorbance of dyes [18]. S depends on the properties of the fabric (substrate) and, thus, the K/S values are directly related to the concentration of dyes on the fabric

Color intensities of the samples

The color intensities of the dyed and acetone-rinsed samples were determined at the wavelength corresponding to the maximum absorbance (K/S_max_) [4]. The colorimetric procedure (outlined above) was repeated 3 times for each sample. The mean of the 3 K/S_max_ values was considered as the color intensity of the sample.

Observation of the dyed fiber surface

A total of 3 types of fabrics (untreated, 30 s plasma-treated and 60 s plasma-treated) after being dyed at 3 different dye concentrations (0.1, 1.0 and 4.5%) were also observed under SEM to observe the effects of the dyeing process on the fiber surface. FESEM (JEOL JSM-7600F) was used for these tests. The condition of SEM was at U = 5 kV, X (magnification) = 5000, 20,000 and 50,000. All the samples were coated with platinum prior to observation by SEM.

## 3. Results

### 3.1. Results of Characterization of the Control and Plasma-Treated Polyester Fabric

#### 3.1.1. The Effect of Plasma Exposure Time on the Tensile Strength of the Woven Polyester Fabric

The tensile strength of the untreated and plasma-treated samples (P_MAX_) are presented in Table 1. The effect of plasma treatment time on the tensile strength of the fabric is expressed as the difference between the tensile strength of the plasma-treated sample compared to the untreated sample in percentage (%). It is calculated using Equation (2).
(2)$$\mathrm{Diference}\;\mathrm{of}\;\mathrm{Pmax}\;\left(\%\right)=\frac{\mathrm{Pmax}\;\mathrm{of}\;\mathrm{untreated}\;\left(N\right)-\mathrm{Pmax}\;\mathrm{of}\;\mathrm{treated}\;\left(N\right)}{\mathrm{Pmax}\;\mathrm{of}\;\mathrm{untreated}\;\left(N\right)}\;100$$

Table 1 shows that all samples had a rather small standard deviation (maximum value only up to 41 N, which is equivalent to 2.5% of the tensile strength). Therefore, these results had the necessary confidence for further analysis.

In Table 1, it can be seen that the differences in the tensile strength of plasma-treated samples compared to the untreated sample were always greater than zero. That means the tensile strength of the treated samples was improved. In practice, the tensile strength in the warp direction of the plasma-treated samples could be considered unchanged since the difference from the untreated sample was lower than the standard deviation. In the weft direction, the tensile strength of the samples treated for 30, 45, 60, 75 and 90 s was slightly increased compared to the untreated samples (from 4.1 to 8.4%). These two trends have both been observed in other papers. In study [17], the tensile strengths in both direction of the polyester fabric were not changed after a DBD plasma treatment with the power of 300 W (1.66 W/cm^2^) for 7 min. Meanwhile, in study [19], the tensile strength of polyester fabric increased by 3.2 and 6.4%, respectively, after RF plasma treatment for 2 and 5 min. This study also suggested that the slight enhancement in the tensile properties might be a result of the plasma etching of the fiber surface which causes the severity of surface roughness, and thus the combined effect of fiber-to-fiber friction due to etching. Study [21] also suggested that a slight increase in the tensile strength of polyester fabric made from spun yarns after atmospheric pressure plasma treatment was the result of an increase in fiber-to-fiber friction due to the etching effect. However, in our study, the tensile strength in the warp direction of all plasma-treated samples did not increase. It is possible that the warp density of fabric was 1.34 times higher than the weft density, resulting in a sufficiently high fiber-to-fiber friction in the warp direction. Therefore, it is possible that the increase in fiber-to-fiber friction due to plasma etching (in the warp direction) was not large enough compared to pre-plasma treatment to produce an increase in the tensile strength of the fabric. Anyway, the exact cause is still not understood.

#### 3.1.2. Effect of Plasma Treatment on the Water Capillary of Fabric

The wicking height of the samples before and after plasma treatment are presented in Table 2 and Table 3.

Table 2 and Table 3 show that wicking height in the warp direction of fabric was clearly improved by the treatment with DBD plasma. It was increased by 15% with a plasma treatment time of just 15 s. When the sample was plasma-treated for 75 or 90 s, their wicking height in the warp direction could be increased up to 22.5%. This phenomenon has also been observed for polyester fabric after low pressure plasma treatment using a mixture of nitrogen, oxygen and argon [21] and after DBD treatment [22]. These phenomena were explained by the association of polar functional groups, as confirmed by the XPS analysis [21,22]. In our study, this phenomenon will also be discussed with the XPS spectra and SEM images of the samples in the next sections.

However, in the weft direction, the wicking heights of the plasma-treated samples were mostly not changed compared to the untreated samples. It is possible that the weft density was too low (44 yarns/inch), which was the reason for the improvement in the capillary capacity being not as obvious as in the warp direction

#### 3.1.3. Effect of Plasma Treatment Time on the Surface of Polyester Fiber

SEM images of the untreated and plasma-treated polyester fibers for 15, 30, 60 and 90 s (×5000 times) are illustrated in Figure 2a,c,e,g,i, respectively. Figure 2b,d,f,h,j show the surface morphology of the untreated polyester fiber and the plasma-treated fiber for 15, 30, 60 and 90 s (at high magnifications).

Figure 2a,b show that the untreated polyester fiber had fairly a smooth surface (excepting for some impurities on the surface), while the surface of all plasma-treated fibers became rough (Figure 2c,e,g,i). However, the surface morphology of the plasma-treated fibers also differed corresponding to the plasma exposure time. Nodules with a size of approximately 100 nm appeared on the plasma-treated fiber surface for 15 s. This phenomenon is similar to that observed in study [4]. However, these nodules were distributed densely on the surface of the plasma-treated sample for 30 s, and the surface of the fiber became completely rough (Figure 2e,f); furthermore, there were protrusions with dimensions greater than 100 nm in this case. This phenomenon was further reinforced for plasma-treated fibers in 60 s (Figure 2g,h). In addition, the fiber surface of the treated fiber for 60 s began to be partially peeled. In particular, this surface destruction tended to produce grooves along the fiber length. Figure 2j shows that after the plasma treatment for 90 s, most of the outermost layer of the polyester fiber was removed. Furthermore, the grooves along the fiber length appeared quite clearly (Figure 2i). This phenomenon was also observed in study [17] when polyester fabric was treated at 300 W for 7 min.

The surface roughness observed in this study explains the increase in the weft tensile strength recorded in Section 3.1.1. The surface roughness and especially the appearance of grooves on the fiber surface may partly explain the increase in wicking height in the warp direction of the fabric (Table 2).

#### 3.1.4. Analysis of X-ray Photoelectron Spectrum (XPS) of the Untreated and Plasma-Treated Samples

XPS spectra of the untreated and plasma-treated samples of polyester are shown in Figure 3. The percentage of chemical groups exhibited on the surface of controlled and plasma-treated polyester are presented in Table 4.

Figure 3a shows three functional groups (C-C/C-H, C-O/C-OH, and O=C-O) on the untreated polyester surface that are characteristic of the polyester structure. Table 4 shows the percentages of these three groups were 39.35, 49.78 and 10.87%, respectively. However, Figure 3b,c show that on the XPS spectra of the plasma-treated samples, a new functional group (O-C-O/C=O) emerged compared to the untreated sample. Furthermore, after plasma treatment, the proportion of functional groups also changed significantly compared to before plasma treatment. The percentage of the C-O/C-OH group decreased by up to 20% after plasma treatment, while the percentage of the O=C-O group and the C-C/C-H group increased by approximately 10%. However, there was no significant difference in the percentage of functional groups when the plasma treatment time increased from 45 to 90 s. This phenomenon is also consistent with the results of the capillary measurements of the samples (it increased by 15% with plasma treatment time in just 15 s, but when the plasma treatment time reached 90 s, the wick height was only 22.5%). It can be said that during plasma treatment some hydroxyl groups of polyester were oxidized to carboxylic groups. As a result, the capillary capacity of the polyester fabric was improved after plasma treatment. This phenomenon was also observed in Study [18]; after the plasma treatment, the percentage of the O=C-O group increased from 10.8 to 18.8%

### 3.2. Effect of Plasma on the Dyeability of the Polyester Fabric

The above results show that immediately after 15 s of plasma treatment, the wick height of the fabric was improved. However, the SEM images show a significant change in fiber surface only in plasma-treated samples for 60 and 90 s. In addition, the percentage of the COOH group of the sample treated for 90 s was not much different from the sample that was treated for only 45 s. For these reasons, as outlined in Section 2.2.3, in this study, only two levels of time (30 and 60 s) were chosen for the plasma treatment of the fabric prior to dyeing.

First, the color uniformity of the dyed samples was assessed visually under D65 light using a light box (STLF, Hanoi, Vietnam). Only samples with a homogeneous color over the entire sample surface were color-measured using a spectrophotometer. The results show that all dyed samples had the required color uniformity for further analysis.

#### 3.2.1. Effect of Plasma Treatment Time on the Color Strength (K/S) Spectra of the Dyed Polyester Fabric

The color strength (K/S) of dyed samples (following the procedure described in Section 2.2.3) given by Ci4200 Spectro Photometer, over the range of 400 to 700 nm are presented in Figure 4.

Figure 4a shows that after 30 and 60 s of plasma treatment, the K/S values of the undyed polyester fabric were virtually unchanged compared to the untreated fabric. 

In general, for red-dyed samples, the reflectance wavelengths were in the range 600 to 700 nm, thus, the K/S of the red color must have been distributed over the remaining wavelength range. As it is seen in Figure 4b–h, the K/S curves of dyed samples show a red color. These figures also show that the K/S values of DBD-treated polyester samples (samples PD 30 and PD 60) increased at the maximum absorption wavelength in range of 450 to 550 nm compared with untreated samples (PD0). Thus, DBD-treated polyester samples appeared to have a darker color than the untreated samples when they were all dyed with the same conditions. This phenomenon has been also observed in the study [17], when the polyester fabric was pretreated with DBD plasma at a power of 1.66 W/cm^2^ for 7 min.

The color intensities of the samples were assessed by their maximum color strength value (K/S_max_). The graphs of K/S values according to the wavelengths of all the dyed samples show that most of the samples had a maximum K/S value at 520 nm (Figure 4b–f). Only the samples dyed with dye concentration of 3 and 4.5% had K/S_max_ at 490 nm (Figure 4g,h). Thus, K/S values at 520 nm of all dyed samples were determined to compare the color strength of all the samples. From these results, three plots showing the relationship between the K/S_max_ values of three types of fabric (PD0, PD30 and PD60) as a function of the dye concentration are shown in Figure 5.

To be able to better understand this phenomenon, SEM images of the dyed samples were also taken to observe the fiber surface after the dyeing procedure (Section 2.2.3). Figure 6 shows the SEM images magnified 5000, 20,000 and 50,000 times of three types of polyester fiber (untreated, plasma-treated for 30 s and plasma-treated for 60 s) after dyeing at dye concentrations of 0.1, 1.0 and 4.5%. The phenomena observed in these images will be discussed with the dyeing behavior of the fabric samples.

Figure 4b–h show that in the range of 450 to 550 nm, most of the plasma-treated samples had a higher K/S_max_ value than those of the untreated samples. However, the difference in K/S_max_ values between the 30 s plasma-treated samples and the untreated samples was not so significant, especially for the dye concentrations of 0.1 and 0.5%. On the other hand, the K/S values of the plasma-treated samples for 60 s were always significantly higher than the K/S values of the untreated samples when they were dyed in the same concentration. This phenomenon appears to be consistent with the dyed fiber surface in Figure 6a,c,e,g,i,k,m,o,q. In these figures, the dye particles were determined by performing SEM at a magnification of 5000 times on the untreated and DBD-treated/dyed polyester fibers. The pictures show that more dye particles were observed on the surface of DBD-treated fabrics. Between two types of DBD-treated samples, more dyes appeared to be observed on the DBD-treated sample for 60 s. A similar phenomena has been observed for disperse dyes in studies [16,17].

Figure 4f–h and Figure 5 show the K/S_max_ values of two fabrics (untreated and DBD-treated for 30 s) appeared to have stabilized at 2 and 3% dye concentrations, suddenly increasing again at concentration 4.5%. Possibly, the rinsing procedure for the dyed sample (under running water (Section 2.2.3)) may not have been able to remove all residual dye for the 4.5% dye concentration.

In Figure 6n (PD0-4.5) and Figure 6p (PD30-4.5), the aggregates of dyes on the fiber surface seemed to be observed. This phenomenon was not observed in other dyed samples. Possibly, they were the surface-bound dye of the dyed samples and they caused the maximum K/S values of the samples dyed at the 3.0 and 4.5% of dyes to move towards the wavelength of 490 nm (Figure 4g,h). In addition, the K/S_max_ values measured in these cases may have been affected by these dye aggregates. To eliminate these suspicions, the color of the dyed fabric samples, which were rinsed off with acetone, will be discussed in the next section.

#### 3.2.2. Effect of Plasma Treatment Time on the Color Strength (K/S) of the Dyed Polyester Fabric/Rinsed with Acetone

K. Gharanjig et al. in study [23], by the measurement of the amount of absorption of two disperse dyes in different solvents, indicated that they were dissolved in acetone and ethanol most probably as monomeric species. In addition, to check the effect of acetone on polyester, the polyester fabric samples before and after plasma treatment for 30 and 60 s were rinsed in acetone according to the procedure outlined in Section 2.2.4. The SEM images of these samples after rinsing with acetone are presented in the Appendix A. These figures show that the surface of the polyester fibers after rinsing for 10 min with acetone appeared to be cleaner than before rinsing (Figure 2). However, the lesions on the fibrous surface due to the plasma etching are still observed in Appendix A. Thus, acetone was chosen as the solvent to wash the dyes off the dyed fabric surface before they were tested for K/S. Furthermore, at room temperature, acetone cannot dissolve the absorbed dyes in the amorphous region of the textile fiber. Therefore, the procedure described in Section 2.2.4 was selected to rinse off the surface-bound dyes of the dyed samples

The dyed samples were rinsed according to this procedure until the acetone after the rinsing cycle was no longer colored. The number of acetone-rinsing cycles performed for different dyed samples was as follows: three cycles for samples dyed with 0.1, 0.5 and 1% of the dye; five cycles for samples dyed with 1.5 and 2% of dyes; 10 cycles for samples dyed with 3% of dye; and 15 cycles for the samples dyed with 4.5% dye. 

The color strength profiles (K/S) of acetone-rinsed samples given by Ci4200 Spectro Photometer, over the range of 400 to 700 nm are presented in Figure 7.

Figure 7a–e show that the K/S curves of samples dyed with a dye concentration less than 2% after acetone rinsing were almost no different from those before acetone rinsing (Figure 4b–f). However, Figure 7f,g show that the K/S curves of the samples dyed with 3 and 4.5% dye concentrations after rinsing with acetone changed compared to before rinsing. While before acetone rinsing, these samples had K/S max values at the wavelength of 490 nm (Figure 4g,h), after rinsing, the maximum K/S values for these samples were determined at 520 nm. Thus, after rinsing with acetone, all dyed samples have K/S _max_ values determined at 520 nm. Furthermore, comparing Figure 7f,g with Figure 4g,h, it can be seen that the K/S max values of the samples after ringsing are higher than before rinsing. To better understand this phenomenon, the color intensity of the samplehơ will be analyzed in the following section.

#### 3.2.3. Color intensities of the dyed and acetone-rinsed samples

As observed in Figure 4 and Figure 6, the K/S values at 520 nm of dyed and acetone-rinsed samples were used as color intensities of the samples. The mean values of the three measurements (following the method presented in Section 2.2.5) and their standard deviation are shown in Table 5. 

Table 5 shows that the standard deviation (SD) value of K/S_max_ value (at 520 nm) of most samples was quite small (less than 10%, only sample PD60’-0.5 had an SD value of approximately 14%). To see more clearly the effect of plasma treatment on the color intensity of the dyed polyester fabrics with the disperse dye, from the results of Table 5, the plots showing the relationship between the K/S values at 520 nm of the dyed fabrics (PD0, PD30 and PD60) and of the dyed/acetone-rinsed fabrics (PD0′, PD30′ and PD60′) as a function of the dye concentration are shown in Figure 8.

The graphs showing the relationship between the K/S values at 520 nm of the rinsed samples (PD’ sets) as a function of the dye concentration are shown in Figure 8 (the bold curves). In a comparison of the K/S max values of the rinsed samples (PD’) with those of before rinsing (PD samples—dashed curves in Figure 8), it can be seen that K/S max values of samples dyed with dye concentrations equal to or less than 2% after rinsing with acetone were almost unchanged compared to before rinsing. However, for samples dyed at 3 and 4.5% dye concentrations, these values were significantly increased compared to before rinsing. This phenomenon was also observed by Ahmed Kerkeni et al. in study [18] when the plasma-treated polyester fabric was dyed with curcumin at 130 °C and then rinsed with ethanol. These authors suggested that when polyester fabric is dyed at concentrations above saturation, the dye molecules are adsorbed onto the fiber surface. Moreover, the physically adsorbed dye molecules could form an ordered monolayer to the multilayer films at the polyester surface, which would reflect light, reducing the K ⁄ S (absorption) values. In addition, Warken et al. [24], who worked on the light absorption of dye molecules deposited on fibers, confirmed that there is absorption reduction as a result of agglomeration of molecules on the fiber surface in an ordered monolayer or multilayer form. Thus, perhaps, before being rinsed off, the dye aggregates observed in Figure 6 affect the K/S values. Therefore, the K/S values of the dyed samples after rinsing with acetone (Three bold curves in Figure 8) were used to evaluate the dyeability of the dyed samples.

The K/S_max_ values of acetone-rinsed samples (three bold curves in Figure 8) show that the color intensities of the plasma-treated/dyed samples increased compared to those of untreated/dyed samples (except for the lowest concentration (0.1%)). Furthermore, plasma treatment time also influenced this difference; the highest color intensity was that of the 60s-plasma-treated/dyed sample, then that of the 30s-plasma-treated/dyed sample, and the lowest was that of the untreated/dyed sample. In addition, the plasma-treated polyester fabric for 60 s almost reached the saturated K/S value at 3.0% of dyes, while the K/S value of the other two fabrics continued to increase at 4.5% concentration. It can be seen that the plasma-treated samples always had a darker color than the untreated samples when they were both dyed under the same conditions. This was particularly significant for samples that were plasma-treated for 60 s. This can allow for dye savings (the K/S value of the 60 s-DBD-treated fabric dyed with 3% dye was almost equivalent to the value of the 30 s-DBD-treated sample dyed with 4.5% dye, and was even higher than the value of the untreated sample dyed with 4.5% dye).

The difference in the dyeing behavior of the plasma-treated samples compared to the untreated samples almost corresponded to the difference in fiber surface morphology between them (Figure 2b,f,h). It is possible that the interactions between the DBD plasma and the material surface changed the microstructure, resulting in an improvement of dyeability. The mechanism of these effects will be discussed in the following section.

## 4. Discussion

In the DBD plasma medium, when the excited and energetic plasma species (ions, radicals, electrons and metastable) are bombarded on to the polyester surface, they initiate various reactions. The first type includes chain scission on the surface which results in surface etching, cleaning or activation [25]. The plasma species, when impacted on the fiber surface, can transfer energy to the polymer substrate. Only the amorphous portion gets degraded and etched away in the initial stage, and the plasma process is such that the electrons and ions attack the amorphous portion [17]. However, the impinging plasma species can transfer enough energy to the polymer matrix to enable the destruction of crystalline domains, which might relax in a disordered amorphous form, causing volume differences at the surface [4]. Thus, the polyester fiber surface can become rough in the same way. In our study, the result of this process was the appearance of nodules approximately 100 nm in size on the fiber surface after 15 s of plasma treatment (Figure 2d). When the plasma treatment time reached 30 s, these nodules could cover the entire fiber surface (Figure 2f). Furthermore, breaking chemical bonds can be caused by plasma etching, leading to the physical removal of molecules [4]. Materials are removed from a polymer surface by physical etching and chemical reactions at the surface to form volatile products [8]. Physical sputtering of materials by chemically non-reactive plasma is a knock-on process by ions with high energy [14]. Chemical etching occurs in chemically reactive types of plasma, and during etching reactions, weight loss of the substrate occurs and the topmost layer of the substrate is stripped off [13]. In our study, this phenomenon was observed in Figure 2h,i,k.

Moreover, beside the bond scission of polymers, the bombardment of ions/electrons in the DBD plasma also can create reactive species on the surface of treatment sample. These reactive species could react with oxygen from the air, resulting in more oxygen-containing polar groups on the polyester surface after the plasma treatment (Figure 3 and Table 4)

The aforementioned physical and chemical modifications on the plasma-treated polyester surface resulted in a higher capillary capacity for the plasma-treated polyester fabric (Table 2).

The improvement in dyeability of the plasma-treated samples (in our study) was also closely related to these modifications. The above morphology modifications could lead to more physical loosening of the microstructure of the fibers, with an increased possibility for the penetration of the dye molecules into the fibers [16]. In addition, an increase in the capillary capacity of the plasma-treated samples also increased the penetration of the dye solution into the fiber structure. The morphological modification of the plasma-treated samples for 60 s was stronger than the samples treated for only 30 s, which is also consistent with their difference in K/S values.

By comparing the dyeing result (Figure 8) with the wicking heights, SEM images and the XPS spectra of the plasma-treated samples, it can be said that the improvement in the dyeability of the plasma-treated polyester fabric was due to the plasma treatment. The plasma treatment caused the physical and chemical surface modification of the material. This lead to more physical loosening of the microstructure of the fibers. This also helped increase the wicking height of the fabric, and it is these modifications that enhance the dyeability of polyester fabrics. When dyeing at the same dye concentration, the color of the plasma-treated fabric was always darker than that of the untreated fabric. However, based on the results of Figure 8, it is possible that the change in the microstructure of the polyester fiber after 60 s of plasma treatment stopped only at the structural loosening of the amorphous region. Thus, the dye absorption of the fabric became easier. However, this has not yet allowed for increased saturated dye absorption of the fabric (it was shown that the maximum K/S value of the samples dyed at 4.5% did not differ significantly).

## 5. Conclusions

The DBD treatment with a power of 1 w/cm^2^ for a short time (from 15 to 90 s) for a polyester fabric (used in the study) could lead to some surface modifications, such as: an increase in the fiber surface roughness and an increase in the number of COOH groups. These surface modifications resulted in an increase in fabric wicking heights in the warp direction up to 22.5%.

Thanks to these positive modifications, the dyability of the polyester fabric with the dispersed dye was enhanced. The improvement in dyeability of the plasma-treated fabric was also dependent on the plasma treatment time. The fabric’s dyeability was significantly increased for the fabric treated with plasma for 60 s. When dyeing at the same dye concentration, the color of the plasma-treated fabric was always darker than that of the untreated fabric. These trends were also observed for fabrics treated with plasma for 30 s; however, with a lower efficiency.

For the fabric used in the study, the aforementioned surface modifications not only did not affect the tensile strength of the fabric, but also increased the tensile strength in the weft direction up to 8.4%. Possibly, an increase in the fiber surface roughness may have resulted in increased friction between the fibers, which was the cause of this increase in tensile strength.

However, the tensile strength in the warp direction and the wicking height in the weft direction of the plasma-treated fabric did not change. It is possible that the difference between the warp density and the weft density of the fabric was the cause of this difference.

Thus, the dyeability of PET fabric with disperse dye can be improved under the conditions plasma power of 1w/cm^2^ in air and treatment time from 30 s. These are quite favorable conditions for the application of plasma in the textile industry on an industrial scale.

However, in this study, the polyester fabric dyeing was carried out only according to the traditional dyeing method (at 130 °C) and color measurements mentioned only the color strength (K/S) of the dyed samples. In order to be able to fully evaluate the impact of the DBD plasma treatment in the short term on the dyeability of polyester fabrics, it is necessary to carry out further dyeing for polyester fabrics at lower temperatures. Furthermore, additional studies on other color properties of dyed fabrics such as color fastness, etc., are also needed. These research contents will be conducted in our next studies.

## Figures and Tables

**Figure 1 polymers-13-01434-f001:**
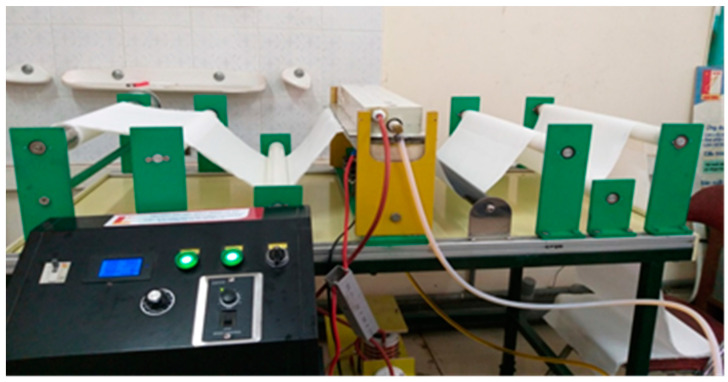
Laboratory DBD plasma equipment.

**Figure 2 polymers-13-01434-f002:**
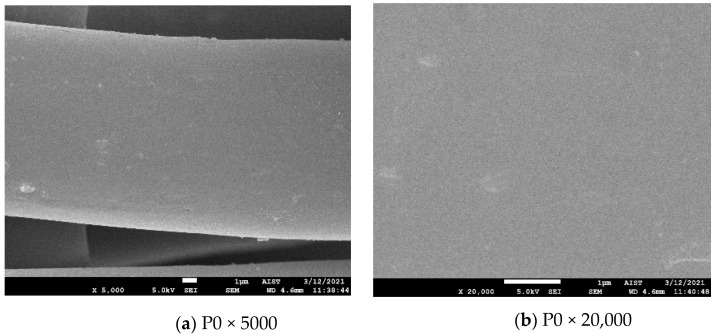
SEM images of the polyester fiber: (**a**,**b**) images of the untreated fiber (P0); (**c**–**i**) and (**j**) images of plasma-treated polyester fibers for 15 (P15), 30 (P30), 60 (P60) and 90 (P90) seconds, respectively.

**Figure 3 polymers-13-01434-f003:**
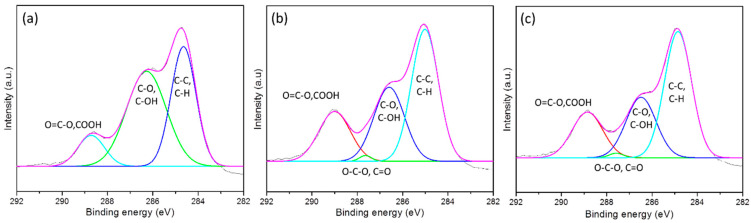
Deconvolution of C1s spectra for (**a**) pristine polyester fabric; (**b**) plasma-treated for 45 s; (**c**) plasma-treated for 90 s.

**Figure 4 polymers-13-01434-f004:**
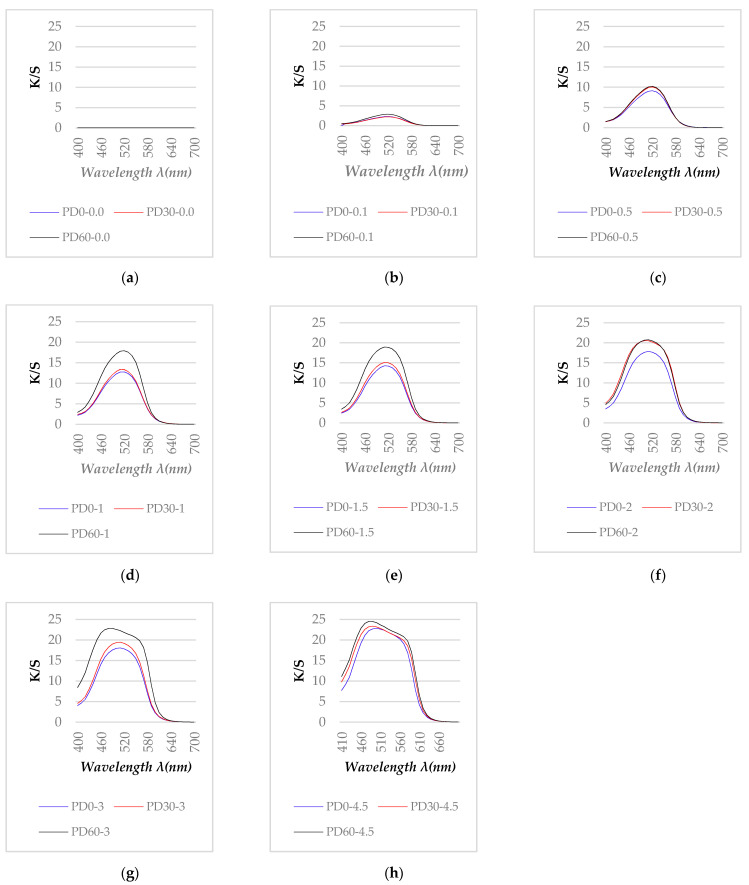
K/S value of untreated and DBD-treated samples before and after dyeing: (**a**) undyed samples (P0, P30 and P60), (**b**) dyed samples with M = 0.1%, (**c**) M = 0.5%, (**d**) M = 1%, (**e**) M = 1.5%, (**f**) M = 2%, (**g**) M = 3%, (**h**) M = 4.5%. In all figures above the red curve is for the untreated polyester/dyed (PD0), blue for the polyester plasma-treated for 30 s/dyed (PD30) and brown for the polyester plasma-treated for 60 s/dyed (PD60).

**Figure 5 polymers-13-01434-f005:**
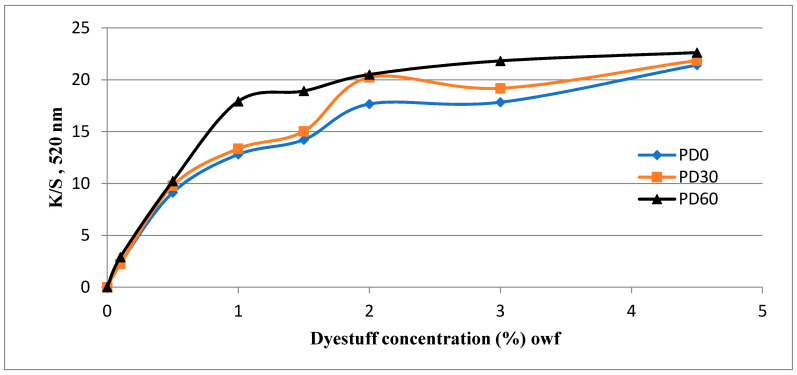
Color strength (K/S at 520 nm) variation of the dyed samples: untreated polyester fabric/dyed (blue curve), DBD-treated polyester fabric for 30 s/dyed (orange curve) and DBD-treated polyester fabric for 60 s/dyed (black curve) as a function of the dye concentration.

**Figure 6 polymers-13-01434-f006:**
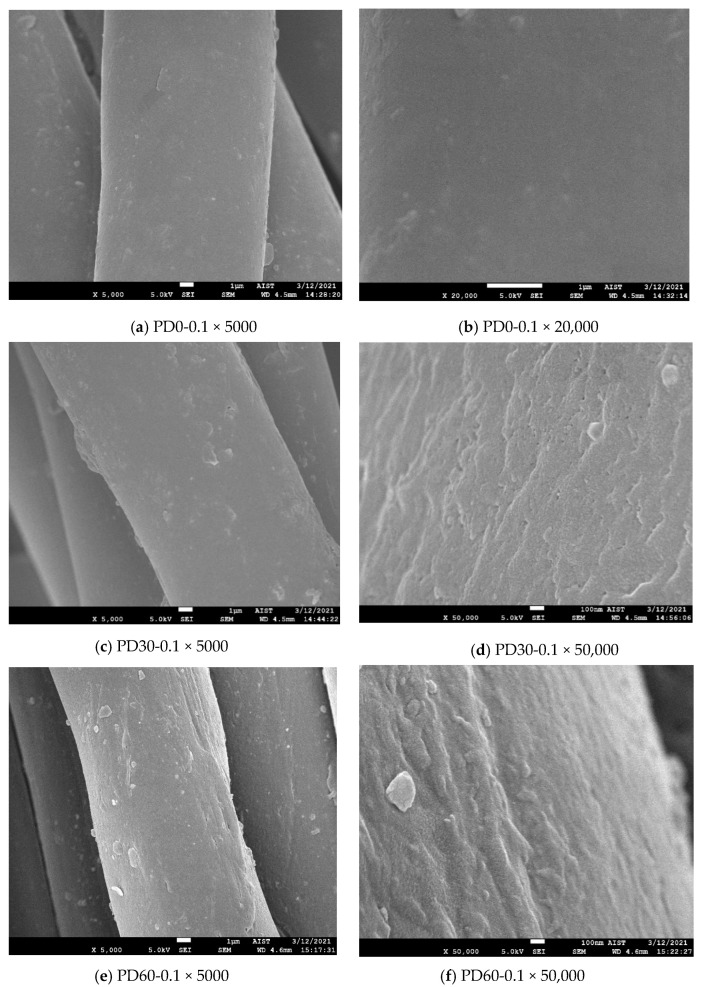
SEM images of the dyed polyester fiber: (**a**,**b**) untreated fiber dyed at 0.1% owf; (**c**,**d**) 30 s plasma-treated fiber dyed at 0.1% owf, (**e**,**f**) 60 s plasma-treated fiber dyed at 0.1% owf, (**g**,**h**) untreated fiber dyed at 1.0% owf, (**i**,**j**) 30 s plasma-treated fiber dyed at 1.0% owf, (**k**,**l**) 60 s plasma-treated fiber dyed at 1.0% owf, (**m**,**n**) untreated fiber dyed at 4.5% owf, (**o**,**p**) 30 s plasma-treated fiber dyed at 4.5% owf; (**q**,**r**) 60 s plasma-treated fiber dyed at 4.5% owf.

**Figure 7 polymers-13-01434-f007:**
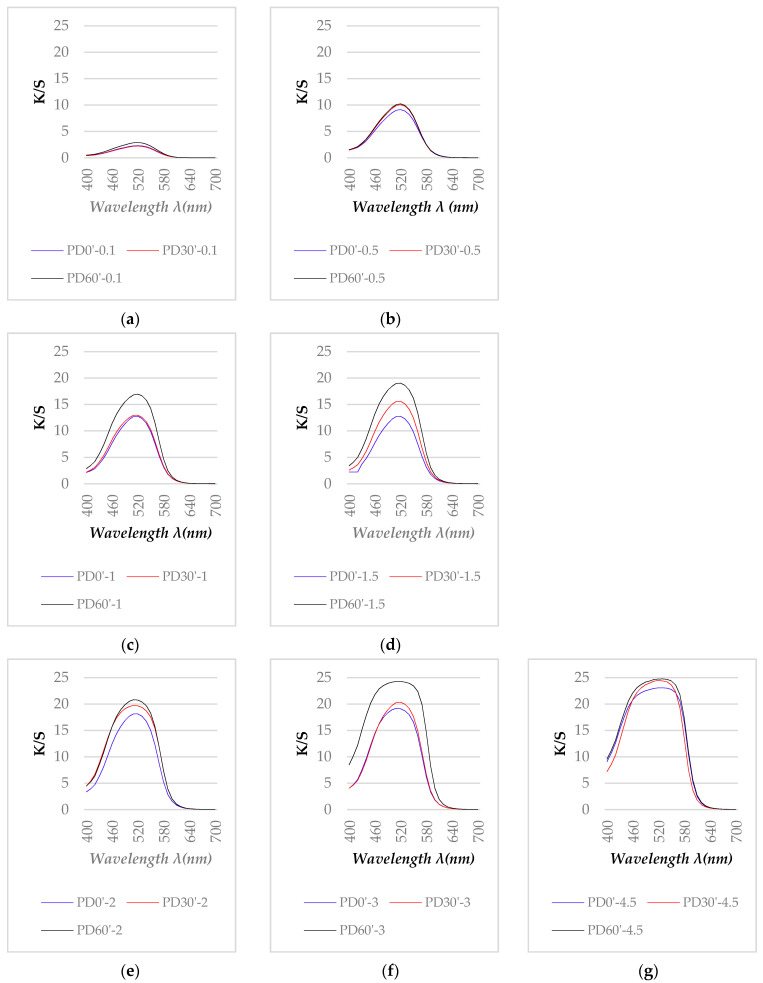
K/S spectra values of three sets of the dyed samples/rinsed in accetone (PD0′, PD30′, and PD60′): (**a**) samples dyed at 0.1% (M = 0.1%), (**b**) M = 0.5%, (**c**) M = 1%, (**d**) M = 1.5%, (**e**) M = 2%, (**f**) M = 3%, (**g**) M = 4.5%. In all figures above, the blue curve is for the untreated polyester (PD0′), red for the plasma treated polyester for 30 s (PD30′) and black for the plasma treated polyester for 60 s. (PD60′).

**Figure 8 polymers-13-01434-f008:**
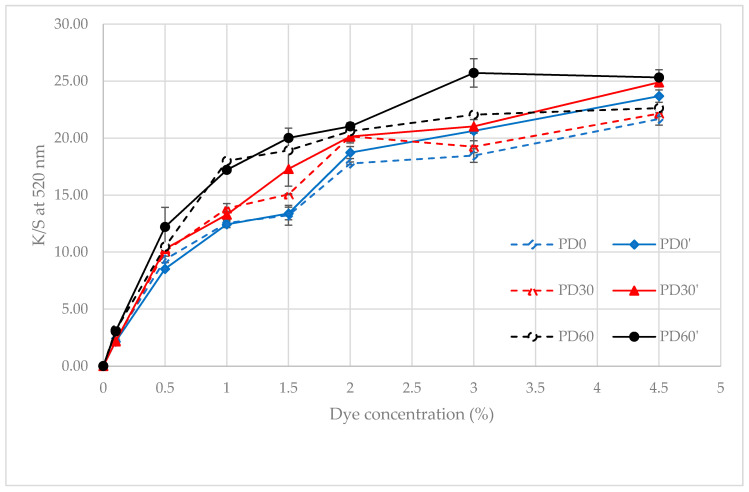
Color intensity (K/S at 520 nm) variation of the dyed samples (PD—dashed curves) and dyed/acetone-rinsed samples with acetone (PD’—bold curves): untreated polyester fabric (blue curves), DBD-treated polyester fabric for 30 s (red curves) and DBD-treated polyester fabric for 60 s (black curves).

**Table 1 polymers-13-01434-t001:** Tensile strength of untreated and treated samples in weft and warp direction.

Plasma Exposure Time (s)	In Warp Direction		In Weft Direction	
	P_MAX_ (N)	Difference of P_MAX_ (%)	P_MAX_ (N)	Difference of P_MAX_ (%)
0	1642 ± 11	-	1059 ± 31	-
15	1648 ± 33	+0.4	1083 ± 31	+2.3
30	1649 ± 20	+0.4	1102 ± 27	+4.1
45	1650 ± 25	+0.5	1123 ± 23	+6.0
60	1651 ± 41	+0.5	1137 ± 27	+7.4
75	1653 ± 16	+0.7	1140 ± 28	+7.6
90	1659 ± 11	+1.0	1148 ± 28	+8.4

**Table 2 polymers-13-01434-t002:** The wicking height of polyester fabric in warp direction.

Plasma Exposure Time (s)	Wicking Height of Samples in Warp Direction (cm) Measured after					
	5 min	10 min	15 min	20 min	25 min	30 min
0	6.3 ± 0.2	8.3 ± 0.1	9.4 ± 0.2	10.0 ± 0.2	10.5 ± 0.1	10.7 ± 0.2
15	6.6 ± 0.3	9.4 ± 0.2	10.8 ± 0.2	11.4 ± 0.2	12.0 ± 0.1	12.3 ± 0.1
30	7.0 ± 0.2	9.5 ± 0.4	10.6 ± 0.1	11.6 ± 0.4	12.2 ± 0.6	12.4 ± 0.5
45	7.4 ± 0.2	8.9 ± 0.3	10.9 ± 0.1	11.5 ± 0.3	12.2 ± 0.2	12.6 ± 0.3
60	7.7 ± 0.3	10.2 ± 0.4	11.3 ± 0.3	12.3 ± 0.3	12.8 ± 0.5	12.5 ± 0.8
75	7.5 ± 0.3	10.5 ± 0.4	11.4 ± 0.4	12.3 ± 0.6	12.6 ± 0.7	13.1 ± 0.7
90	7.5 ± 0.1	10.4 ± 0.1	11.5 ± 0.2	12.1 ± 0.3	12.6 ± 0.2	13.1 ± 0.2

**Table 3 polymers-13-01434-t003:** The wicking height of PES fabric in weft direction.

Plasma Exposure Time (s)	Wicking Height of Samples in Weft Direction (cm) Measured after					
	5 min	10 min	15 min	20 min	25 min	30 min
0	5.9 ± 0.1	7.9 ± 0.1	9.1 ± 0.2	9.7 ± 0.2	10.4 ± 0.3	10.6 ± 0.2
15	5.8 ± 0.3	7.6 ± 0.3	8.2 ± 0.4	9.1 ± 0.2	9.6 ± 0.3	10.0 ± 0.4
30	5.9 ± 0.1	7.9 ± 0.1	9.1 ± 0.1	9.7 ± 0.2	10.4 ± 0.1	10.7 ± 0.2
45	6.1 ± 0.1	8.1 ± 0.1	9.2 ± 0.2	10.0 ± 0.2	10.4 ± 0.2	10.9 ± 0.3
60	6.7 ± 0.1	8.4 ± 0.2	9.4 ± 0.2	10.0 ± 0.2	10.4 ± 0.2	10.6 ± 0.1
75	6.0 ± 0.2	7.9 ± 0.2	8.9 ± 0.2	9.6 ± 0.2	10.1 ± 0.3	10.3 ± 0.3
90	5.7 ± 0.2	7.5 ± 0.1	8.5 ± 0.1	9.3 ± 0.2	9.7 ± 0.2	10.0 ± 0.2

**Table 4 polymers-13-01434-t004:** Percentage of the chemical groups present on the surface of pristine and plasma-treated polyester.

Plasma Exposure Time for the Samples (s)	Percentage of Functional Groups on the Surface of the Polyester Fibers (%)			
	C-C/C-H	C-O/C-OH	O-C-O/C=O	O=C-O
0	39.35	49.78	-	10.87
45	47.62	30.40	0.93	21.05
90	51.00	27.06	0.95	20.98

**Table 5 polymers-13-01434-t005:** K/S value at 520 nm of untreated and DBD-treated polyester fabrics after dyeing (PD0, PD30 and PD60) and after dyeing/acetone-rinsing (PD0′, PD30′ and PD60′).

Dye Concentration (%)	Type of Sample					
	PD0	PD0′	PD30	PD30′	PD60	PD60′
0	0.01 ± 0.00	0.01 ± 0.00	0.01 ± 0.00	0.01 ± 0.00	0.01 ± 0.00	0.01 ± 0.00
0.1	2.38 ± 0.07	2.15 ± 0.11	2.21 ± 0.02	2.17 ± 0.04	3.16 ± 0.23	3.05 ± 0.01
0.5	9.36 ± 0.28	8.53 ± 0.06	10.07 ± 0.07	10.28 ± 0.23	10.49 ± 0.23	12.21 ± 1.72
1	12.56 ± 0.25	12.44 ± 0.28	13.84 ± 0.41	13.29 ± 0.29	18.00 ± 009	17.21 ± 0.23
1.5	13.23 ± 0.87	13.39 ± 0.56	15.07 ± 0.04	17.30 ± 1.51	18.93 ± 0.10	20.02 ± 0.86
2	17.78 ± 0.12	18.73 ± 0.53	20.15 ± 0.57	20.14 ± 0.42	20.61 ± 0.11	21.02 ± 0.33
3	18.47 ± 0.60	20.63 ± 1.34	19.25 ± 0.52	21.03 ± 0.60	22.06 ± 0.21	25.72 ± 1.25
4.5	21.69 ± 0.54	23.68 ± 0.54	22.15 ± 0.26	24.89 ± 0.40	22.64 ± 0.15	25.32 ± 0.68

## Data Availability

Not applicable.

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
