# Peer review of "The Effect of DBD Plasma Activation Time on the Dyeability of Woven Polyester Fabric with Disperse Dye"

_polymers, 2021, doi:10.3390/polym13091434_

Round 1
Reviewer 1 Report
1. Line 245:What is the meaning of “PES fabrics”? Also line 250, and so on.
2. Line 249, Table 2: Only the results of 30 mins DBD-treated fabrics were added the average error? Also Table 3 and others. Besides, the number should be kept in consistence in that of containing an estimated number, such as 7 should be 7.0, 10 → 10.0, and 6 → 6.0, and so on.
3. Line 276: In my opinion, Figure 1 can be deleted, which was repeated with those results in Tables 2 and 3.
4. Line 342-343: The authors should confirm the accuracy of the writing of sample name, such as PD0-3 appears corresponding to dye concentration of 3% and 2% (Table 5), also PD30-3 (Table 6), PD60-3 (Table 7).
5. Line 354: The K/Sλmax was much lower than that of the untreated fabric above the dye concentration of 1% when the plasma treatment time was 60 s. Why? There was no explanation about this.
6. Line 165-170: Whether should the writing of color strength be K/Sλmax or K/Smax? They are different in the graph and figure caption.
7. Line 391: The “it” should be “It”.
8. In my opinion, Table 9 can be deleted.
9. The authors should discuss more about the true mechanism of the effects of the plasma treatment on the resulted performance of the fabrics such as the improved dyeability. Although the wettability and roughness were improved, how did they affect the dyeability or dyeing process? Do not just discuss the results, but more theoretical or scientific analysis.
Author Response
Thank you very much for your helpful comment. The details of the revisions according to each comment of reviewer 1 are as follows:
- Line 245:What is the meaning of “PES fabrics”? Also line 250, and so on.
That our mistake. We change all to Polyester.
- Line 249, Table 2: Only the results of 30 mins DBD-treated fabrics were added the average error? Also Table 3 and others. Besides, the number should be kept in consistence in that of containing an estimated number, such as 7 should be 7.0, 10 → 10.0, and 6 → 6.0, and so on.
We did check the data and added the average error as well as kept the estimated number in table 2,3
- Line 276: In my opinion, Figure 1 can be deleted, which was repeated with those results in Tables 2 and 3.
Yes, We keep the table and deleted the figure.
- Line 342-343: The authors should confirm the accuracy of the writing of sample name, such as PD0-3 appears corresponding to dye concentration of 3% and 2% (Table 5), also PD30-3 (Table 6), PD60-3 (Table 7).
Yes, The writing of sample name having mistake. We did change them.
- Line 354: The K/Sλmaxwas much lower than that of the untreated fabric above the dye concentration of 1% when the plasma treatment time was 60 s. Why? There was no explanation about this.
We have re-measure the K/S of all samples , and show the data in figure 4,5
- Line 165-170: Whether should the writing of color strength be K/Sλmaxor K/Smax? They are different in the graph and figure caption.
We did changed to K/Smax.
- Line 391: The “it” should be “It”.
Sorry, we could not find the line 391 in our manuscript
- In my opinion, Table 9 can be deleted.
Yes, we total agree. The Table 9 have been deleted
- The authors should discuss more about the true mechanism of the effects of the plasma treatment on the resulted performance of the fabrics such as the improved dyeability. Although the wettability and roughness were improved, how did they affect the dyeability or dyeing process? Do not just discuss the results, but more theoretical or scientific analysis
Yes, we adding an discussion part and explain more in this part. (lines 637 – 716)

Reviewer 2 Report
The manuscript writing by Kim-Thu Nguyen Thi et al. studies the effect of DBD plasma treatment on the dyeability of woven polyester fabric. The authors show that the dyeability of the fabric was enhanced by plasma treatment (ca. 90s), meanwhile, the tensile strength of the fabric was retained. A publication may be suggested, but the following comments/questions should be addressed:
- The authors should explain at the molecular-level why plasma treatment can enhance the dyeability of polyester. A scheme on the changes of the chemical structure of polyester after plasma and its interactions with dye molecules should be offered in the MS.
- In the abstract “The results show that the longer the plasma treatment time, the higher the wettability of the polyester fabric”. From which data, the authors made such conclusion, at least the values of water contact angle should be offered.
- The plasma treatment is just surface sensitive. It’s hard to understand that a plasma treatment (especially in air) in 30-90s could cause “nodules of a size less than 1µm appearing on the fiber surface” (Line 1 on Page 9). The authors should explain the reasons.
- How does the plasma treatment influence the stability and durability of dyes on polyester?
- SEM images after dyeing should be offered in comparison to Figure 2.
Author Response
Thank you very much for your helpful comment. The details of the revisions according to each comment of reviewer 1 are as follows:
- The authors should explain at the molecular-level why plasma treatment can enhance the dyeability of polyester. A scheme on the changes of the chemical structure of polyester after plasma and its interactions with dye molecules should be offered in the MS.
Yes, we have been explained more in the discussion section (lines 637 – 716)
- In the abstract “The results show that the longer the plasma treatment time, the higher the wettability of the polyester fabric”. From which data, the authors made such conclusion, at least the values of water contact angle should be offered.
Sorry, the sentence is not correct. We already re-write the abstract
- The plasma treatment is just surface sensitive. It’s hard to understand that a plasma treatment (especially in air) in 30-90s could cause “nodules of a size less than 1µm appearing on the fiber surface” (Line 1 on Page 9). The authors should explain the reasons.
We already take new SEM photo with better equipment. The image was presented in figure 2d. In this image, we confirm that the nodules approximately 100 nm.
- How does the plasma treatment influence the stability and durability of dyes on polyester?
In this study, we did not study on the stability and durability of dyes on polyester. We hope that this issue can be discussion in our future work.
- SEM images after dyeing should be offered in comparison to Figure 2.
Yes, the photo of samples which dyed at concentration of 0.1; 1.0; 4.5 have been taken, and we did show that results in to Figure 6.

Reviewer 3 Report
The authors investigated the dyeability of woven polyester fabrics upon DBD plasma activation time pretreatment. The plasma activation involved air instead of pure gasses and exposure times were shorter then reported previously. The effect of pretreatment on fabrics samples was investigated by scanning electron microscopy, photoelectron spectroscopy, tensile strength and wicking height measurements. The dyeability (K/S) of plasma treated woven samples improves overall with (1) plasma discharge time and (2) dyestuff concentration when compared to the untreated samples.
I have 2 concerns regarding the presentation of Fig. 4: (1) The data (Table 8) must be included in Fig. 4. (2) The equations used to fit the data have no physical grounds. By including the data in Fig. 4, the reader can judge by him/herself the quality of fit. Equations for which k/s=0 for dye conc=0.0% and k/s reaches saturation values for high dye conc. might give better fits and have a more physical foundation.
The authors might consider other ways to convince the reader of dyeability improvement by plotting the measured (not fitted) K/S improvement versus dye concentration for DS30 and DS60 in one figure.
My comments are as follows:
1) The authors correctly state that the wicking height is 15% to 22% higher for the treated fabrics in the warp direction. But they should also mention that in the weft direction all 6 samples show a decrease for all wicking heights (5mins to 30 mins) after 90 seconds plasma exposure time.
2) In Fig. 1b I detected an error for the data point (P60seconds,30 minutes)=10.6. In Fig. 1(b) this datum is located at ~10.3.
3) The K/Smax values in Table 8 were fitted for PD0, PD30 and PD60 using the function Y=aln(x)+const. What is the physical reason to use this equation. For dye concentration reaching infinity the K/Smax would become unrealistically high and for low dye concentrations below ~0.07% K/Smax becomes negative. The assumed fitting equations therefore seem not sufficient reliable for use in fabric dyability analysis as stated on line 375.
4) Please include the data points from Table 8 in Fig. 4. Then the reader can judge by eye how well the equations fit.
5) Fig. 8: fitting equations for which at 0%dye conc. results in K/S=0 and that for high dye conc. approaches asymptotic values might give higher correlation coefficients. For example the function Y(%conc)=Ymax(1-exp(-%conc/tconc). This function results in K/Smax(0)=0 and K/Smax(inf conc)=Ymax. The authors might verify such kind of equation.
6)The color pictures in Tables 6, and 7 appear convincingly “more deeper/intense colored” for similar dyestuff concentrations as compared to the untreated samples. The authors might consider other ways to convince the reader the dyeability improvement by plotting the K/S improvement versus dye concentration for DS 30 and DS 60 in one figure. Then it becomes clear that for conc>=1.5% 7 of the 8 samples K/S improved (of which 5 are very convincing). For 60s plasma exposure always the best results are obtained (conc>=1.5%) when compared with treated and 30s treated samples.
7) Nearly all references in ref. list do not provide the journal in which the work was published, coauthor names are also missing. Please provide for each reference the required information.
8) Various minor errors were detected such as: line 182: 6x109Torr; line 245: damages should be damage; 90 (should be 90 seconds); line 259 (should be Fig. 1a); line 309: polyster should be polyester. Line 391 it is these should be it are these. Please go carefully through the whole text
9) Lines 242-243: “In Study [19], whether the treatment time was 2 minutes or 5 minutes, the tensile strength did not change compared to the untreated sample.” This sentence is not correct since a 6.5% increase in tensile strength was measured in Ref.[19] (Heibesh et al.) between untreated and 5 minute treated samples. Heibesh et al. noted in ref.[19] from their data in Table 2 a slight increase of tensile strength with plasma exposure time: 31 Kgf (untreated); 32 Kgf (2 minutes) 33 Kgf (5 minutes) and mentioned as possible explanation an increase in fibre surface roughness resulting in fibre to fibre friction due to etching.
Therefore the correct sentence should be something like “In Study [19], whether the treatment time was 2 minutes or 5 minutes, the tensile strength did not change much (increased be less then 7%) compared to the untreated sample.”
10) In Conclusions the authors mention that tensile strength is not affected. But actually it increases by 8.4 % in weft direction (Table 1).
Author Response
Thank you very much for your helpful comment. The details of the revisions according to each comment of reviewer 1 are as follows:
The authors investigated the dyeability of woven polyester fabrics upon DBD plasma activation time pretreatment. The plasma activation involved air instead of pure gasses and exposure times were shorter then reported previously. The effect of pretreatment on fabrics samples was investigated by scanning electron microscopy, photoelectron spectroscopy, tensile strength and wicking height measurements. The dyeability (K/S) of plasma treated woven samples improves overall with (1) plasma discharge time and (2) dyestuff concentration when compared to the untreated samples.
I have 2 concerns regarding the presentation of Fig. 4: (1) The data (Table 8) must be included in Fig. 4. (2) The equations used to fit the data have no physical grounds. By including the data in Fig. 4, the reader can judge by him/herself the quality of fit. Equations for which k/s=0 for dye conc=0.0% and k/s reaches saturation values for high dye conc. might give better fits and have a more physical foundation.
Thank you very much. We total agree and the equation that we did used to fit the data have been removed. We re-draw the K/S chart and the data are included in the chart (figure 5, 8)
The authors might consider other ways to convince the reader of dyeability improvement by plotting the measured (not fitted) K/S improvement versus dye concentration for DS30 and DS60 in one figure.
Thank you very much, we did it in figure 5 and 8
My comments are as follows:
1) The authors correctly state that the wicking height is 15% to 22% higher for the treated fabrics in the warp direction. But they should also mention that in the weft direction all 6 samples show a decrease for all wicking heights (5mins to 30 mins) after 90 seconds plasma exposure time.
Thank you, We have corrected it in section 3.1.2. (line 343 – 345 and 357- 361)
2) In Fig. 1b I detected an error for the data point (P60seconds,30 minutes)=10.6. In Fig. 1(b) this datum is located at ~10.3.
We did review the data and removed the figure 1b.
3) The K/Smax values in Table 8 were fitted for PD0, PD30 and PD60 using the function Y=aln(x)+const. What is the physical reason to use this equation. For dye concentration reaching infinity the K/Smax would become unrealistically high and for low dye concentrations below ~0.07% K/Smax becomes negative. The assumed fitting equations therefore seem not sufficient reliable for use in fabric dyability analysis as stated on line 375.
Thank you very much. The equation is not correct. We have removed it.
4) Please include the data points from Table 8 in Fig. 4. Then the reader can judge by eye how well the equations fit.
Yes, we did that in figure 4 and figure 8.
5) Fig. 8: fitting equations for which at 0%dye conc. results in K/S=0 and that for high dye conc. approaches asymptotic values might give higher correlation coefficients. For example the function Y(%conc)=Ymax(1-exp(-%conc/tconc). This function results in K/Smax(0)=0 and K/Smax(inf conc)=Ymax. The authors might verify such kind of equation.
We review the data and did not use this equation anymore.
6)The color pictures in Tables 6, and 7 appear convincingly “more deeper/intense colored” for similar dyestuff concentrations as compared to the untreated samples. The authors might consider other ways to convince the reader the dyeability improvement by plotting the K/S improvement versus dye concentration for DS 30 and DS 60 in one figure. Then it becomes clear that for conc>=1.5% 7 of the 8 samples K/S improved (of which 5 are very convincing). For 60s plasma exposure always the best results are obtained (conc>=1.5%) when compared with treated and 30s treated samples.
We have added the rinsing step with acetone after the dyeing process. then the K/S at 520 nm of the dyed samples after rinsing with acetone as function of dye concentration (Fig.8) can show the improvement in dyeing of plasma treated samples.
7) Nearly all references in ref. list do not provide the journal in which the work was published, coauthor names are also missing. Please provide for each reference the required information.
Thank you, all references have changed to the right format.
8) Various minor errors were detected such as: line 182: 6x109Torr; line 245: damages should be damage; 90 (should be 90 seconds); line 259 (should be Fig. 1a); line 309: polyester should be polyester. Line 391 it is these should be it are these. Please go carefully through the whole text
Yes, we did change them.
9) Lines 242-243: “In Study [19], whether the treatment time was 2 minutes or 5 minutes, the tensile strength did not change compared to the untreated sample.” This sentence is not correct since a 6.5% increase in tensile strength was measured in Ref.[19] (Heibesh et al.) between untreated and 5 minute treated samples. Heibesh et al. noted in ref.[19] from their data in Table 2 a slight increase of tensile strength with plasma exposure time: 31 Kgf (untreated); 32 Kgf (2 minutes) 33 Kgf (5 minutes) and mentioned as possible explanation an increase in fibre surface roughness resulting in fibre to fibre friction due to etching.
Therefore the correct sentence should be something like “In Study [19], whether the treatment time was 2 minutes or 5 minutes, the tensile strength did not change much (increased be less then 7%) compared to the untreated sample.”
Thank you very much. We have changed this sentence.
10) In Conclusions the authors mention that tensile strength is not affected. But actually it increases by 8.4 % in weft direction (Table 1).
Thanks, we adding more discussions.

Round 2
Reviewer 2 Report
My concerns have been addressed by the authors in the revision. A publication of the current form can be suggested.
